# Secure Ring Signature Scheme for Privacy-Preserving Blockchain

**DOI:** 10.3390/e25091334

**Published:** 2023-09-14

**Authors:** Lin Wang, Changgen Peng, Weijie Tan

**Affiliations:** 1State Key Laboratory of Public Big Data, College of Computer Science and Technology, Guizhou University, Guiyang 550025, China; gs.linwang20@gzu.edu.cn (L.W.); wjtan@gzu.edu.cn (W.T.); 2Key Laboratory of Advanced Manufacturing Technology, Ministry of Education, Guizhou University, Guiyang 550025, China

**Keywords:** blockchain, ring signature, privacy protection, distributed key generation, elliptic curve cryptography

## Abstract

Blockchain integrates peer-to-peer networks, distributed consensus, smart contracts, cryptography, etc. It has the unique advantages of weak centralization, anti-tampering, traceability, openness, transparency, etc., and is widely used in various fields, e.g., finance and healthcare. However, due to its open and transparent nature, attackers can analyze the ledger information through clustering techniques to correlate the identities between anonymous and real users in the blockchain system, posing a serious risk of privacy leakage. The ring signature is one of the digital signatures that achieves the unconditional anonymity of the signer. Therefore, by leveraging Distributed Key Generation (DKG) and Elliptic Curve Cryptography (ECC), a blockchain-enabled secure ring signature scheme is proposed. Under the same security parameters, the signature constructed on ECC has higher security in comparison to the schemes using bilinear pairing. In addition, the system master key is generated by using the distributed key agreement, which avoids the traditional method of relying on a trusted third authorizer (TA) to distribute the key and prevents the key leakage when the TA is not authentic or suffers from malicious attacks. Moreover, the performance analysis showed the feasibility of the proposed scheme while the security was ensured.

## 1. Introduction

Blockchain technology is the underlying technology of Bitcoin, a concept mentioned by Nakamoto [1] in his Bitcoin white paper published in 2008, which has triggered new industrial and technological revolutions and is a popular research area at present. In essence, blockchain is a kind of distributed database [2], and its underlying chain structure is the data blocks arranged in chronological order, which inherits the technologies of smart contracts [3], peer-to-peer networks (P2P) [4], the consensus mechanism [5], cryptography, etc., has the unique advantages of weak centralization, being tamper-proof, traceability, openness, transparency, etc., and is able to realize the direct circulation of value between untrusted nodes without the need for third-party institutions, which not only reduces the trust cost of transactions, but also greatly shortens the interaction time. It is considered as a key technology to realize the transformation of the “information Internet” to the “value Internet” [6].

Blockchain’s technical advantages make it applicable to various domains. One of them is digital currencies, where Bitcoin and its derivatives are expanding fast. According to a 2017 report by ARK Investments, there are more than 10 million Bitcoin users worldwide, with over USD 150-million in daily transactions [7]. In the financial sector, blockchain technology is highly valued by central banks, who design their own digital currencies by applying or studying it. They use blockchain technology to enhance the traditional financial system, which suffers from long delays and low efficiency in reconciliation, clearing, and cross-border settlement, as well as the high costs of maintaining central ledger data. In healthcare, the companies and stakeholders are benefiting from blockchain technology, which helps them streamline business processes, improve patient outcomes, manage patient data, comply with regulations, cut costs, and leverage healthcare-related data more effectively [8]. In the energy sector, blockchain technology, which is based on decentralization, is seen as a game-changing technology for creating distributed energy systems. It offers a decentralized trust mechanism that can be used for distributed energy operations and can help overcome management weaknesses and challenges in distributed energy systems [9]. In the field of cultural industry, the data inerrancy and high trustworthiness of the blockchain industry can be utilized to carry out many businesses such as certificate storage, digital property rights protection, and cultural relic identification. Furthermore, there are other fields such as justice, military, and supply chains that are gradually using blockchain to improve the problems existing in each field.

As blockchain technology continues to develop and become widely used, the privacy leakage concerns confronting it are becoming increasingly noticeable and must be given sufficient attention. While the blockchain mechanism avoids potential failures of individual servers and the exposure of data in terms of data storage, all transactions have to be made public to all nodes in order to reach consensus among the decentralized blockchain nodes. This exposes the privacy of the transactions greatly, and a main challenge is the safeguarding of users’ identity privacy. Identity privacy refers to the connection between a user’s real identity and the blockchain nodes. The blockchain stores data in an unchangeable way as a distributed ledger that any node can access. Transactions on the blockchain are somewhat anonymous, but not completely secure against privacy breaches. With advanced computing techniques, an attacker can track and analyze the correlation of public data in a global ledger to reveal sensitive information. For example, if there are consistent and correlated transactions, an attacker can extract some user characteristics using a graph of transactions between different addresses [10]. Moreover, an attacker can search all possible transactions to obtain the transaction addresses and estimate the balances, which can help to infer the user’s identity and location [11]. Cryptography is the prevalent method for privacy protection among researchers. Therefore, it should be combined with blockchain technology to offer a suitable solution for the blockchain’s privacy issue and guarantee the safety of users’ data.

A common security technique in blockchain systems is to use digital signatures to check the integrity of a document or a message. This ensures non-repudiation. The ring signature scheme [12] is a special type of group signature [13], which uses a group of *L* in a user’s private key and *L* in all the users’ public keys to complete the signature; the signature of the verifier can only verify that the signature comes from the group, but who signed the name is unknown. The ring signature does not need a trusted center and can hide the identity of the signer, which protects the user’s privacy. Considering its features, the scheme can be applied to anonymous payment applications or untraceable transactions and also to other scenarios that require privacy protection, such as elections, voting, and identity verification. Therefore, we propose a new privacy protection scheme by combining the ring signature algorithm and blockchain technology. However, the key generation link of traditional ring signature algorithms requires a secret key management center, which will face the risk of key leakage and increase the probability of attackers forging ring signatures. Moreover, most of the ring signatures are constructed based on bilinear pairing, and the computational complexity is usually high, which may lead to longer signing and verification time. It also lacks security. In order to improve the efficiency and security of ring signatures, we also made a new design of the scheme.

To solve the above-mentioned issues in the blockchain system, this paper studied and analyzed the related ring signature schemes and designed a new ring signature scheme suitable for the blockchain environment by using the anonymity feature of the ring signature. The scheme is based on Shamir verification secret sharing theory [14] and the Feldmanprotocol [15]; the key generation link in the scheme was improved, and the main body of the scheme adopted the elliptic curve cryptography principle.

The main contributions of the paper are stated as follows:The algorithm exploits the concept of distributed key generation to create a system master key, which enhances the process of distributing the key by a trusted authorizer (TA) in traditional signature algorithms and eliminates the risk of key leakage when the TA is untrustworthy or subjected to malicious attacks.This scheme is a ring signature constructed based on ECC, which provides better security with the same length of key compared to the scheme based on bilinear pairing. This algorithm strengthens the signature’s unforgeability, which reduces the attackers’ probability of succeeding in cracking the key.This scheme improves the efficiency of ring signature generation and verification and is more compatible with the environment of blockchain systems.

The rest of this paper is organized as follows. In Section 2, we review different blockchain privacy-preserving schemes. In Section 3, we provide some preliminary concepts, including general ring signature algorithms and security model definitions. Section 4 describes the algorithms and system models of secure ring signature schemes. Section 5 proves the security of the proposed scheme, and Section 6 gives the efficiency and performance comparison of the signature schemes. Finally, the paper is summarized in Section 7.

## 2. Related Works

Blockchain technology faces challenges in its development in existing industries due to privacy breaches and other security issues. To protect users’ identity privacy in blockchain, different schemes have been suggested to increase the anonymity level. In order to resist the book analysis technique, researchers propose a defense mechanism for exchanging assets and obfuscating addresses, i.e., the address obfuscation mechanism, in response to the assumptions on which the technique is based. In 2014, Bonneau et al. [16] proposed the Mixcoin protocol, which enhances asset security through an electronic-signature-based commitment mechanism. In 2015, Valenta et al. [17] proposed the Blindcoin protocol, which guarantees the internal privacy of centralized hybrid coin schemes by using blind signature techniques. In 2018, Ziegeldorf et al. [18] proposed CoinParty; it occupies a unique position in the design space of hybrid services by decrypting the novel combination of hybrid nets and threshold signatures, combining the advantages of previously proposed centralized and decentralized hybrid services into a single system. The address obfuscation mechanism can protect the privacy of the ledger to a certain extent; however, the result of address obfuscation will still be stored in the public ledger, and an attacker can threaten user privacy to a certain extent by analyzing the obfuscated transactions with features.

Ledger information hiding mainly preserves the confidentiality of the ledger by encrypting the private data in the ledger and provides “credentials” through cryptographic techniques to keep the correctness of the blockchain ledger verifiable. Most of the existing techniques for the implementation of the ledger-information-hiding mechanism belong to zero-knowledge proof techniques [19], which means that zero-knowledge proofs do not convey any proof of knowledge other than the correctness of the proposition under discussion. Li et al. [20], based on the ring-zero-proof-of-knowledge and blockchain technology, proposed a secure and efficient fair transaction mechanism for a sharing environment. The mechanism utilizes ring-zero-knowledge proofs to hide transaction contents and relationships without affecting authentication by adding a new trusted player. However, zero-knowledge proofs have a high computational cost and storage space, require the use of complex cryptographic algorithms and a large number of data structures, and may affect the performance and scalability of blockchain systems.

Ring signature schemes allow participants to sign messages with the group name and preserve the confidentiality of the signer’s identity from disclosure. The ring signature technique has two main features: first, any member of the group can issue the correct signature alone; second, any member of the group only knows whether he or she has initiated the signature, and members outside the group only know whether the signer belongs to the group. A higher level of privacy is achieved by VOTOR, a practical remote voting scheme that uses product anonymization channels and linkable ring signatures. It was proposed by Thomas et al. [21]. An information sharing system that ensures the confidentiality of applicants was created by Patil et al. [22] using an ID-based ring signature technique. They removed the certificate validation process to make the system secure and reliable. A new ring signature scheme based on elliptic curves was proposed by Li et al. [23], which improves the signature unforgeability and anonymity compared with the traditional ring signature scheme. Wang et al. [24] proposed a flexible threshold ring signature scheme in chronological order, which in practice has the advantage of solving both the update problem and the chronological order problem. In addressing the difficulty of sharing medical records between healthcare organizations, Lai et al. [25] introduced a secure medical-data-sharing scheme based on traceable ring signatures and blockchain. Samra et al. [26] proposed a new framework, a certificate-less aggregation scheme based on traceable ring signatures (CLA-TRS), which ensures conditional privacy-preserving authentication in vehicular ad-hoc network (VANET) communications. Table 1 summarizes the application scenarios, techniques, advantages, etc., of the above schemes. It can be seen that most of the above schemes are based on bilinear pairing construction, and their security needs to be improved. Moreover, most of them rely on a trusted key generation center (KGC), which cannot avoid attacks and reduces the difficulty of forging signatures.

## 3. Preliminaries

In this context, we will present some preliminary knowledge including elliptic curves, difficult assumptions, and the general ring signature algorithm and its security models below.

### 3.1. Elliptic Curve

An elliptic curve is not an ellipse; it is called an elliptic curve because the equation for the curve is similar to the equation for calculating the perimeter of an ellipse. In general, the curve equation of an elliptic curve is a cubic equation of the following form:$$y^{2}+axy+by=x^{3}+cx^{2}+dx+e,$$
where a,b,c,d, and *e* are real numbers satisfying some simple conditions.

Elliptic curves over finite fields are commonly used in cryptography, which refers to the curve defined by Equation (1) in which all coefficients are elements in a finite field GF(q), where *q* is a large prime number. The most-commonly used of these is the curve defined by Equation (Equation 2):$$y^{2}\;mod\;q=\left(x^{3}+ax+b\right)\;mod\;q,$$
where a,b∈GF(q) and $△=(4a^{3}+27b^{2})$ mod q≠0.

An elliptic curve is symmetric with respect to the X-axis, and the addition operation on it is defined as follows: if three points lie on the same line, the sum of them is *O*. Addition on an elliptic curve is defined as follows:*O* is the additive identity element, that is, for any point *P* of the elliptic curve, P+O=P.Let $P_{1}=\left(x,y\right)$ be a point on an elliptic curve whose additive inverse element is defined as $P_{2}=P_{1}=\left(x,y\right)$. This is because, when the connection of $P_{1}$ and $P_{2}$ is extended to infinity, another point *O* on the elliptic curve is obtained, that is the three points $P_{1},P_{2}$, and *O* on the elliptic curve are collinear, so $P_{1}+P_{2}+O=O$, $P_{1}+P_{2}=O$, that is $P_{2}=-P_{1}$.Let $P=\left(x_{p},y_{p}\right),Q=\left(x_{q},y_{q}\right)$ and P≠−Q, then $R=P+Q=(x_{r},y_{r})$ is determined by the following rule:
(1)$$x_{r}\equiv \left(\lambda^{2}-x_{p}-x_{q}\right)\;mod\;q,$$
(2)$$y_{r}\equiv \left(\lambda^{2}\left(x_{p}-x_{r}\right)-y_{p}\right)\;mod\;q,$$
where
(3)$$\lambda =\left\{\begin{array}{c} \frac{y_{q}-y_{p}}{x_{q}-x_{p}}\;mod\;q,\;if\;P\neq Q\\ \frac{3x_{p}^{2}+a}{2y_{p}}\;mod\;q,\;if\;P=Q \end{array}\;\right.$$The multiple of a point *P* is defined as 2P=P+P.

### 3.2. Problem Assumptions

**Definition** **1.**
*Elliptic curve discrete logarithm problem (ECDLP): Given any two points P, Q on an elliptic curve $E(F_{p})$, solving for the value x satisfying the equation Q=x·P is unsolvable in polynomial time.*


### 3.3. Ring-Signature-Generation Algorithm

The ring signature is a unique type of group signature that uses a set of public keys instead of one. It hides the identity of the actual signer from the verifier. Unlike other group signatures, ring signatures do not require a manager or any coordination among the members. The basic ring signature has three components: KeyGen(), Sign(), and Verify():KeyGen(): This algorithm needs to input a security parameter *l* and, then, generate a key pair (pk,sk) for each user, where pk is the public key and sk is the private key.Sign(): This algorithm takes the message *m*, which needs to be encrypted, the private key sk of a ring member, and the public key set $L=\{pk_{1},pk_{2},\cdots ,pk_{n}\}$ of the selected ring members and generates a signature σ for the message *m*. One of the parameters in the signature σ follows a ring according to certain rules.Verify(): This algorithm is a deterministic algorithm, which takes the public key set $L=\{pk_{1},pk_{2},\cdots ,pk_{n}\}$, the message *m*, and the signature σ as the input and outputs “accept” if the verification passes and “reject” otherwise.

### 3.4. Security Models

The ring signature scheme is supposed to satisfy the requirements of correctness, unconditional anonymity, and unforgeability.

#### 3.4.1. Game I Correctness

The output of the ring-signature-generation algorithm serves as the input for the ring signature verification algorithm, which always outputs acceptance. The unforgeability and unconditional anonymity of a ring signature scheme are defined by a game between a simulator R and an adversary A. To begin with, we introduce A’s inquirable oracle machines JO, CO, and SO:Join oracle machine (JO(⊥) →PK): With this query, a new user is added to the system and the public key PK of the new user is output.Corruption oracle machine (CO($PK_{i}$) →$sk_{i}$: The user’s public key $PK_{i}$ is input, and the corresponding private key $sk_{i}$ is output.Signed oracle machine (SO(m,n,L,PK) →σ): Input signed message *m*, and set $L=\{PK_{1},PK_{2},\cdots ,PK_{n}\}$ of public keys of size *n*; the signer’s public key $PK_{\pi }$(1≤n≤π) returns a valid ring signature σ.

The definition of the general ring signature and the ring signature defined in this paper contains four basic algorithms: system initialization algorithm, key-generation algorithm, ring signature generation, and ring signature verification. The key point is that the general ring signature puts forward more-specific requirements for the signature-value-generation process: Given a message M, the public key ($PK_{1},PK_{2},\cdots ,PK_{n}$) of *n* members, the signer’s private key $sk_{\pi }\left(1\leq \pi \leq n\right)$, and a secure hash function, produce a set $r_{1},r_{2},\cdots ,r_{n},h_{1},h_{2},\cdots ,h_{n}$, and finally, output the signature value σ, where $r_{i}\neq r_{j},i\neq j,h_{i}\left(1\leq i\leq n\right)$ are the hash values determined by $m,r_{i}\left(1\leq i\leq n\right)$ and the public keys of the ring members, the signature value σ is completely determined by $r_{1},r_{2},\cdots ,r_{n},h_{1},h_{2},\cdots ,h_{n}$, and message *m* is decided.

#### 3.4.2. Game II Unforgeability

The unforgeability of a ring signature is defined by the following game between a simulator R and an adversary A:R generates the system parameters params and sends them to A.A adaptively queriesoracles JO, CO, and SO and random oracles H.A outputs a signature message $M^{*}$, a set $S^{*}$ consisting of *n* user public keys, and two forged signature values $\sigma_{0}^{*}$, $\sigma_{1}^{*}$.

A is said to have won the above game if the following four conditions are met:

*Step 1*: $\sigma_{0}^{*}$, $\sigma_{1}^{*}$ are valid ring signatures on the message $M^{*}$, that is *RVerify*($M^{*}$, $S^{*}$, $\sigma_{i}^{*}$), (i∈{0,1}) → accept.

*Step 2*: All public keys in $S^{*}$ are obtained by querying the oracle JO.

*Step 3*: All the public keys in $S^{*}$ are not corrupted, that is the adversary cannot obtain the private keys of any ring member in $S^{*}$.

*Step 4*: $\sigma^{*}$ is not obtained by querying the signed oracle machine SO.

A ring signature scheme is said to be unforgeable if, for any *PPT* adversary A, the probability of winning the above game is negligible.

#### 3.4.3. Game III Unconditional Anonymity

The unconditional anonymity of a ring signature scheme is defined by a game between a simulator R and an adversary A with infinite computational power as follows:R generates the system parameters params and sends them to A.A can adaptively query join oracle machine JO.A sends a signature message $M^{*}$ and a set $S^{*}=PK_{1},PK_{2},\cdots ,PK_{n}$ consisting of public keys of *n* users to R, where all public keys are obtained by the JO query. R randomly selects π∈{1,2,⋯,n} and computes a signature $\sigma_{\pi }=Sign\left(M^{*},n,S^{*},sk_{\pi }\right)$, where $sk_{\pi }$ is the private key corresponding to $PK_{\pi }$. Finally, R sends $\sigma_{\pi }$ to A.A outputs a guess $\pi^{\prime }\in \left\{1,2,\cdots ,n\right\}$.

A ring signature is said to satisfy unconditional anonymity if, for an adversary A with infinite computing power, the probability of guessing the correct signer π is at most $\frac{1}{n}$, where *n* is the cardinality of the public key set *S*.

## 4. Secure Ring Signature Scheme

In this section, a secure blockchain ring signature scheme is proposed by incorporating the idea of distributed key generation. The following is a detailed description of the system model and the signature algorithm.

### 4.1. System Description

The system model is shown in Figure 1 and contains entities such as the group users, the distributed generation center KDC, and the blockchain network. The purpose of the KDC here is to generate the system parameters, validate and manage the cluster personnel, and verify the signature. The specific scheme is described as follows:

*Step 1*: The KDC picks the security parameter *l* and generates the system parameters for the signature.

*Step 2*: The user $A_{\pi }$ applies to the KDC to become a member of the group; $A_{\pi }$ sends $ID_{\pi }$ to the KDC; the KDC passes the verification, returns $Q_{\pi }$, and adds the user $A_{\pi }$ to the group; the user completes the registration.

*Step 3*: The member $A_{\pi }$ interacts with other group members $A_{i}$ (i=1,2,⋯,n,i≠π) through a secure channel to generate the system’s master private key and master public key.

*Step 4*: The member $A_{\pi }$ signs the data message *m* to generate signature $\sigma_{\pi }$.

*Step 5*: All KDCs in the blockchain system verify the signature $\sigma_{\pi }$ and upload the data information and signatures to the blockchain database after verification.

Miner nodes in the blockchain network pack the set of transactions for a period of time and then continuously calculate the random numbers that meet the conditions to construct blocks that meet the predefined conditions for confirming the transactions. The KDCs mentioned in this paper can be considered as miner nodes, also known as full nodes, and users as regular nodes. Full nodes act as servers in the distributed network, and they maintain consensus rules among other nodes, as well as transaction validation. Ordinary nodes retain some of the information on the block. After the group users sign the data information, all KDCs verify the signatures and upload them into the blockchain network.

### 4.2. Algorithm Description

#### 4.2.1. Setup Algorithm

The system server selects security parameters *l* and randomly picks a large prime number q>l. *G* is a base point on the elliptic curve. Let $G_{1}$ be an additive group of order *q* generated by the generating element *P*. The hash functions are: $H_{0}:{\left\{0,1\right\}}^{*}\to E\left(F_{q}\right),H_{1}:{\left\{0,1\right\}}^{*}\to Z_{q}^{*},H_{2}:{\left\{0,1\right\}}^{*}\times G_{1}\to Z_{q}^{*},H_{3}:{\left\{0,1\right\}}^{*}\to G_{1}$. *N* represents the number of authorized users; $\left\{q,G,G_{1},H_{0},H_{1},H_{2},H_{3},N\right\}$ is the public parameter. When the system parameters are determined, every authorized user $A_{i},\left(i=1,2,\cdots N\right)$ picks a random polynomial of degree N−1, $f_{i}\left(x\right)=a_{i0}+a_{i1}x+a_{i2}x^{2}+\cdots +a_{i(N-1)}x^{\left(N-1\right)}$ over $Z_{q}^{*}$, where $f_{i}\left(0\right)=a_{i0}$. Each $A_{i}$ computes and broadcasts $T_{ij}=P^{a_{ij}}\left(mod\;q\right)\left(j=0,1,\cdots ,N-1\right)$. Meanwhile, every authorized user $A_{i}$ transmits the calculated secret value $s_{ik}=f_{i}\left(k\right)$ through a secure channel to the other $A_{k}=\left(k=0,1,2,\cdots ,N-1,k\neq i\right)$ in the group. After that, every $A_{k}$ receives the secret value $s_{ik}$ and determines if it is correct by using the formula $P^{s_{ik}}\overset{?}{=}\prod_{j=0}^{N-1}{\left(T_{ij}\right)}^{i^{j}}\left(mod\;q\right)$. If the equation is not satisfied, the secret value is wrong, and $A_{k}$ sends an error message to $A_{i}$, who has to resend the right secret value until the equation holds. Then, the master key $S=\sum_{i=1}^{N-1}b_{i0}\left(mod\;q\right)$ and the master public key $P_{0}=S\cdot P$ are established by the *N* authority members.

#### 4.2.2. Key Generation

For each authorized signer of the system, $A_{i}$ transmits its identity information $ID_{i}$ to the KDC. Then, the KDC randomly selects $k_{i}\in Z_{q}^{*}$, computes $Q_{i}=H_{3}(ID_{i}\left|\right|k_{i})$, and secretly transmits it to $A_{i}$ over a secure channel. Next, signer $A_{i}$ computes $D_{i}=S\cdot Q_{i}$, randomly picks $x_{i}\in Z_{q}^{*}$, and computes its private key $sk_{i}=H_{2}\left(x_{i}\cdot D_{i}\right)$ and public key $pk_{i}=sk_{i}*G$.

#### 4.2.3. Signature Generation

Assuming that the signature user in the system is π, the public key is $pk_{\pi }=sk_{\pi }*G$, and the private key is $sk_{\pi }=H_{2}\left(x_{\pi }\cdot D_{s}\right)$. Choose a set $L=\{ID_{1},ID_{2},\cdots ,ID_{n}\}$ consisting of *n* identities of other authorized users of the system, and if the public key $pk_{\pi }$ of the system is not in *L*, assign the attribute values $Y_{i},A_{i}$ for each public key $pk_{i}$ as follows:

*Step 1*: Randomly chose $v_{i},t_{i},r_{i}\in Z_{q}^{*}$, and compute:(4)$$Y_{i}=\left\{\begin{array}{cc} (v_{i}+t_{i})*G, & if\;i=\pi \\ \left(t_{i}+r_{i}\right)*pk_{i}+v_{i}*G & if\;i\neq \pi \end{array}\right.$$(5)$$A_{i}=\left\{\begin{array}{cc} \left(v_{i}+t_{i}\right)*H_{0}\left(pk_{i}\right), & if\;i=\pi \\ v_{i}*H_{0}\left(pk_{i}\right)+\left(t_{i}+r_{i}\right)*I_{\pi } & if\;i\neq \pi \end{array}\right.$$
where: $I_{\pi }=sk_{\pi }*H_{0}(pk_{\pi })$ is a message signature image that prevents double-spending attacks in the system. It is obtained by mapping $pk_{i}$ to a curve point in the finite field using $H_{0}\left(pk_{i}\right)$.

*Step 2*: Randomly select $s\in Z_{q}^{*}$, and then, calculate:(6)$$h=H_{2}\left(m||s\right).$$(7)$$\begin{array}{c} c_{i}=\left\{\begin{array}{cc} H_{1}\left(h,Y_{1},\cdots ,Y_{n},A_{1},\cdots ,A_{n}\right)-\sum_{i=1,\;i\neq \pi }^{n}c_{i} & if\;i=\pi \\ t_{i}+r_{i} & if\;i\neq \pi , \end{array}\right. \end{array}$$(8)$$\begin{array}{c} d_{i}=\left\{\begin{array}{cc} \left(v_{i}+t_{i}\right)-c_{i}*sk_{i} & if\;i=\pi \\ v_{i}, & if\;i\neq \pi \end{array}\right. \end{array}$$
where: *m* stands for the content of the signature, and the final output of the transaction initiator π’s ring signature for the message *m* is $\sigma =(I_{\pi },c_{1},c_{2},\cdots ,c_{\pi },\cdots ,c_{n},d_{1},d_{2},\cdots ,d_{\pi },\cdots ,d_{n})$.

#### 4.2.4. Verify

The following steps can be used to verify the transaction signature σ by anyone who possesses the public keys of all the members in the ring signature.
(9)$$\left\{\begin{array}{c} \zeta_{i}=c_{i}*pk_{i}+d_{i}*G\\ \eta_{i}=c_{i}*I_{\pi }+d_{i}*H_{0}\left(pk_{i}\right) \end{array}\right.$$
(10)$$\sum_{i=1}^{n}c_{i}=H_{1}\left(h,\zeta_{1},\zeta_{2},\cdots ,\zeta_{n},\eta_{1},\eta_{2},\cdots ,\eta_{n}\right)$$

Compute $\zeta_{i},\eta_{i}$ using Equation (Equation 9), and check if Equation (Equation 10) holds. If it does, the signature image $I_{\pi }$ in the signature is not used, and the signature is valid. If it does not, the signature image $I_{\pi }$ is used and the signature is invalid.

## 5. Security Analysis

### 5.1. Correctness Analysis

The verifier checks the transaction signature σ using Equation (Equation 10), and if it holds, the signature is valid. When i≠π, the conversion of $\zeta_{i}$ is given as Equation (Equation 11) and $\eta_{i}$ is given as Equation (Equation 12):(11)$$\begin{array}{cc} \zeta_{i} & =c_{i}*pk_{i}+d_{i}*G\\ \; & =\left(t_{i}+r_{i}\right)*pk_{i}+v_{i}*G\\ \; & =Y_{i} \end{array}$$
(12)$$\begin{array}{cc} \eta_{i} & =d_{i}*H_{0}\left(pk_{i}\right)+c_{i}*I_{\pi }\\ \; & =v_{i}*H_{0}\left(pk_{i}\right)+\left(t_{i}+r_{i}\right)*I_{\pi }\\ \; & =A_{i} \end{array}$$

When i=π, the conversion of $\zeta_{i}\;,\eta_{i}$ is as follows:(13)$$\begin{array}{cc} \zeta_{i} & =c_{i}*pk_{i}+d_{i}*G\\ \; & =c_{i}*pk_{i}+\left[\left(v_{i}+t_{i}\right)-c_{i}*sk_{i}\right]*G\\ \; & =v_{i}*G+t_{i}*G\\ \; & =Y_{i} \end{array}$$
(14)$$\begin{array}{cc} \eta_{i} & =d_{i}*H_{0}\left(pk_{i}\right)+c_{i}*I_{\pi }\\ \; & =\left[\left(v_{i}+t_{i}\right)-c_{i}*sk_{i}\right]*H_{0}\left(pk_{i}\right)+c_{i}*sk_{\pi }*H_{0}\left(pk_{\pi }\right)\\ \; & =v_{i}*H_{0}\left(pk_{i}\right)+t_{i}*H_{0}\left(pk_{i}\right)\\ \; & =A_{i} \end{array}$$

Therefore, based on the above relationship, the validity of our proposed scheme can be verified according to the following equation.
(15)$$\begin{array}{cc} \; & H_{1}\left(h,\zeta_{1},\zeta_{2},\cdots ,\zeta_{n},\eta_{1},\eta_{2},\cdots ,\eta_{n}\right)\\ \; & =H_{1}\left(h,Y_{1},Y_{2},\cdots ,Y_{\pi },\cdots ,Y_{n},A_{1},A_{2},\cdots ,A_{\pi },\cdots ,A_{n}\right)\\ \; & =c_{\pi }+\sum_{i=1,i\neq \pi }^{n}c_{i}\\ \; & =\sum_{i=1}^{n}c_{i} \end{array}$$

### 5.2. Unforgeability Analysis

**Theorem** **1.**
*Under a randomized oracle model, the messages m can be chosen adaptively by adversary A in Game II to attack. If there exists an algorithm that can win the ECDLP game in polynomial time T, then it is shown that the ECDLP hard problem can be broken with a non-negligible probability.*


**Proof.** The purpose of challenger R is to compute the value of *a* when provided with a random instance of the discrete logarithm problem (P,aP). The challenger R sets the public key of the signer $U_{*}$ as: $pk_{i^{*}}=aP$. In this scenario, R acts as a subroutine within A and takes on the role of the challenger in Game II. To simplify the discussion, let us assume that all queries made by the attacker A are distinct. Next, we elaborate on how the challenger R deals with A’s query:*Step 1*, **initialization**: The challenger R proceeds to execute the initialized algorithm with a security parameter of *l* to obtain the system parameters. Subsequently, these system parameters are transmitted from R to the adversary A.*Step 2*, **hash query**: This step consists of the challenger R creating an empty table *L*, where *L* holds pairs of two values, such as $\left(x_{i},y_{i}\right)$, where the challenger R randomly selects $y_{i}$ and sets $H_{1}\left(x_{i}\right)=y_{i}$. When the adversary A queries $H_{1}\left(x_{i}\right)$, R hands over $y_{i}$ to A and appends $\left(x_{i},y_{i}\right)$ to list *L*.*Step 3*, **public key query**: When the adversary A queries the public key of a user, the challenger R halts if $sk_{i}=sk_{i^{*}}$; otherwise, the challenger R gives the matching user public key $pk_{i}$ to adversary A.*Step 4*, **private key query**: When the adversary A queries the private key of a user, if $pk_{i}=pk_{i^{*}}$, then R stops operating, in the absence of this, the R sends the appropriate user private key $sk_{i}$ back to adversary A.*Step 5*, **ring signature query**: The adversary A transmits information *m* and a public key collection *L* of *N* users to R, which returns a corresponding signature σ. Suppose there is a user identity $pk_{\pi }\in L$ such that $pk_{s}\neq pk_{i^{*}}$, then the adversary A signs the message using $pk_{\pi }$ as the real signer and gives the signature σ. Alternatively, the adversary A will conduct the next steps:
Randomly choose $v_{i},t_{i},r_{i},s\in Z_{q}^{*}$, and compute:
(16)$$Y_{i}=\left\{\begin{array}{cc} (v_{i}+t_{i})*G, & if\;i=\pi \\ v_{i}*G+\left(t_{i}+r_{i}\right)*pk_{i^{*}} & if\;i\neq \pi \end{array}\right.$$
(17)$$A_{i}=\left\{\begin{array}{cc} \left(v_{i}+t_{i}\right)*H_{0}\left(pk_{i^{*}}\right), & if\;i=\pi \\ v_{i}*H_{0}\left(pk_{i^{*}}\right)+\left(t_{i}+r_{i}\right)*I_{\pi }^{*} & if\;i\neq \pi \end{array}\right.$$
(18)$$h=H_{2}\left(m||s\right),$$
(19)$$\begin{array}{c} c_{i}=\left\{\begin{array}{cc} H_{1}\left(h,Y_{1},\cdots ,Y_{n},A_{1},\cdots ,A_{n}\right)-\sum_{i=1,i\neq \pi }^{n}c_{i} & if\;i=\pi \\ t_{i}+r_{i} & if\;i\neq \pi \end{array}\right. \end{array}$$
(20)$$\begin{array}{c} d_{i}=\left\{\begin{array}{cc} \left(v_{i}+t_{i}\right)-c_{i}*sk_{i^{*}} & if\;i=\pi \\ v_{i}, & if\;i\neq \pi \end{array}\right. \end{array}$$The ring signature is given as $\sigma^{*}=\left(I_{\pi }^{*},c_{1},c_{2},\cdots ,c_{\pi }^{*},\cdots ,d_{1},d_{2},\cdots ,d_{\pi }^{*},\cdots ,d_{n}\right)$.
*Step 6*, **forgery**: At last, the adversary A provides the signer $pk_{i^{*}}$ with a signature with different information $m^{*}$. This same result can be obtained by the challenger R, while both signatures σ and $\sigma^{*}$ are valid: $\sigma =(I_{\pi },c_{1},c_{2},\cdots ,c_{\pi },\cdots ,d_{1},d_{2},\cdots ,d_{\pi },\cdots ,d_{n})$, $\sigma^{*}=\left(I_{\pi }^{*},c_{1},c_{2},\cdots ,c_{\pi }^{*},\cdots ,d_{1},d_{2},\cdots ,d_{\pi }^{*},\cdots ,d_{n}\right)$.The challenger R returns the value corresponding to the private key $a=sk_{\pi }$.Hence, the adversary A for the instantiation (P,aP) can be found for $a=sk_{\pi }$, which means that the ECDLP is solved. □

Assuming that A can forge valid ring signatures with a non-negligible probability, there exists an algorithm R that addresses the ECDLP in polynomial time. However, the ECDLP is known to be hard, so the probability of forging the ring signatures in our scheme will be negligible with a random oracle model. For the ring signature $\sigma =(I_{\pi },c_{1},c_{2},\cdots ,c_{\pi },\cdots ,d_{1},d_{2},\cdots ,d_{\pi },\cdots ,d_{n})$, the adversary needs to obtain the signer’s private key in the computation even if the adversary randomly selects $v_{i},t_{i},r_{i}$ to forge $Y_{i},A_{i},Y_{\pi },A_{\pi }$, and $d_{\pi }$. In the proposed scheme, the user’s public and private keys are calculated from the system’s master key pair, which is jointly generated by the authorized user group according to the distributed key-generation algorithm and is not issued by the trusted third party. Therefore, there is no third-party attack, so the user’s public and private keys are safe. Without knowing the key, it is infeasible to compute the key image $I_{\pi }=sk_{\pi }H_{0}\left(pk_{\pi }\right)$, so an attacker cannot create the signature σ. Therefore, the scheme in this paper is unforgeable.

### 5.3. Unconditional Anonymity

**Theorem** **2.**
*In the signature scheme proposed in this paper, the signer has unconditional anonymity, i.e., for any algorithm T, any participant ensemble $L=pk_{1},pk_{2},\cdots ,pk_{n}$, and any $pk_{\pi }\in L$, the probability $P_{r}\left[pk=pk^{\prime }\right]$ is always $\frac{1}{2}$, where the signer of π creates a ring signature: $\sigma =(I_{\pi },c_{1},c_{2},\cdots ,c_{\pi },\cdots ,d_{1},d_{2},\cdots ,d_{\pi },\cdots ,d_{n})$.*


**Proof.** Step 1: The challenger R computes the system parameters and gives them to the adversary A.Step 2: The adversary A performs polynomially restricted ring signature queries adaptively.Step 3: The adversary A outputs a message *m*, two different public keys $pk_{1},pk_{2}$ selected from the set of public keys *L* consisting of authorized users in the challenge phase, and delivers all of this information to R. Next, R randomly chooses one of the two public keys to generate the ring signature and transmits the ring signature $\sigma =(m,L,sk_{u})$ to the adversary A.Step 4: The adversary A performs polynomially restricted ring signature queries adaptively.Step 5: Finally, the adversary A gives a public key $pk^{\prime }\in \left\{0,1\right\}$.Step 6: The adversary A succeeds in this game if and only if $pk=pk^{\prime }$.The output signature cannot be seen by any third party until the signer has voluntarily disclosed all information himself/herself. In the ring signature generation, the signer computes the $Y_{i}$ and $A_{i}$ values needed to obtain $c_{i}$ and $d_{i}$ by randomly picking the corresponding $t_{i},v_{i},r_{i}\in Z_{q}^{*}$ and also obtains the private key by randomly choosing $x_{i}\in Z_{q}^{*}$ and computing $sk_{i}=H_{2}\left(x_{i}\cdot D_{i}\right)\in Z_{q}^{*}$. Therefore, the ring signature σ is uniformly distributed in *G*. The chance that a non-member can guess the real signer is at most 1/(n+1), and the chance that a member of the ring group can guess the real signer is at most 1/n, so the ring signature scheme meets unconditional anonymity. □

## 6. Performance Evaluation

In this section, the computational efficiency of the secure ring signature scheme based on distributed key generation proposed in this paper is analyzed. To achieve a credible security level, we adopted the experiment that has been performed for the computation evaluation in [25]. The experimental environment was: an i58500CPU@3.00GHz, 8GBRAM on an HP desktop, based on the Windows 10 operating system, under the Eclipse development environment, using JAVA Version 1.8.0 and JPBC Version 2.0.0 for the implementation, which uses the library Type A class curves to construct symmetric prime-order bilinear groups and performs Type A pairing on the super-singular elliptic curve E. The equation $y^{2}\equiv x^{3}+x$ mod *p* defines *E*, where p≡3 mod 4, the embedding degree is two, and the order of $G_{1}$ is *q*. The order of the group is 512 bit, and the order of the Galawa domain is 160 bit. The signatures of our scheme in the cryptographic operations can be found in Table 2. We define the execution time of some notations of the cryptography operations in milliseconds (ms) in Table 3.

In our proposed scheme, the user signature requires the ECC-based scalar multiplication operation, a hash operation mapping to points on elliptic curves, a multiplication operation, and a one-way hash operation mapping to a finite field of prime numbers, with the last one having negligible computational overhead. Therefore, the computational overhead of generating a signature is:$$T_{Sign}=\left(4n-2\right)T_{SM_{E}}+\left(2n-1\right)T_{H_{P}}+T_{M}$$

The signature verification process of this program mainly requires the ECC-based scalar multiplication operation. Thus, the computational cost is:$$T_{Verify}=4nT_{SM_{E}}$$

The signature communication cost is (2n+1)L, where *L* is the bit length of the group $G_{1}$, based on the signature-generation phase of our scheme. This shows that the signature length increases linearly with the number of users.

To conclude, the time consumption of the three different ring signature schemes in the signature-generation and signature-verification steps is summarized in Table 4. It was observed that our scheme took less time than the other schemes, both in the signature-generation phase and the signature-verification phase. This indicates that our signature scheme possesses higher signing and verification efficiency.

## 7. Conclusions

To address the privacy leakage problem faced by users in blockchain systems, this paper proposes a ring signature scheme suitable for blockchain. The design of the scheme is based on the elliptic curve cryptography and distributed key generation ideas. First of all, by generating the system master key through the DKG, the risk of key leakage when a trusted authorizer (TA) is attacked can be effectively reduced. Furthermore, the security analysis of the ring signature showed that the scheme enhances the unforgeability and anonymity of the signature. Under the same security parameter length, the elliptic-curve-based ring signature design is more secure than the traditional bilinear pairing design. Finally, by comparing with related signature schemes, our scheme had a shorter signature generation and verification time and higher efficiency. In view of the problem of the efficiency and communication overhead of the scheme increasing with the increase of the users, we will study the aggregation scheme of the ring signature in the future to improve the verification and communication efficiency.

## Figures and Tables

**Figure 1 entropy-25-01334-f001:**
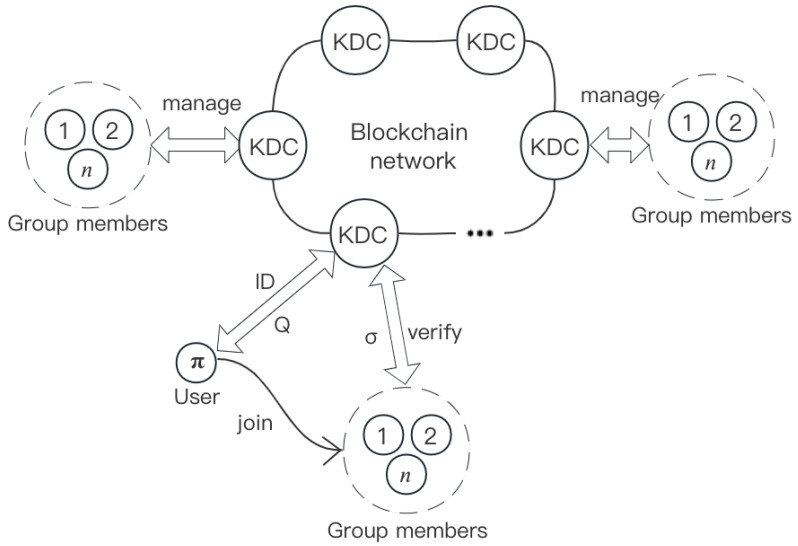
System model.

**Table 1 entropy-25-01334-t001:** Comparison of ring signature schemes.

Scheme	Scenario	Techniques	Advantages	Drawbacks
[21]	Vote	Bilinear Hash Anonymous- channel	Linkable Practical	Relies on trusted center Lack of efficiency analysis
[22]	Cloud computing	Bilinear Hash ID-based	Simplified management High efficiency	Relies on trusted center Does not support key revocation and update
[23]	Blockchain	ECC Hash	Improves unforgeability Improves anonymity	Lack of efficiency analysis Relies on trusted center
[24]	Edge computing	Bilinear Hash Threshold	Flexible Renewable	Relies on trusted center Lack of efficiency comparison of related schemes
[25]	Medical sharing	Bilinear Hash DKG	Traceable Controllable	High computational cost
[26]	VANET	Bilinear Hash ECC	Traceable High efficiency	Relies on trusted center

**Table 2 entropy-25-01334-t002:** Notations of cryptography operation.

Notation	Crypto-Operation
$SM_{E}$	ECC-based scalar multiplication operation.
$A_{E}$	ECC-based point addition operation.
$H_{P}$	Map-to-point operation.
*H*	One-way hash function operation, which is negligible.
*M*	Multiplication operation.
*P*	Bilinear pair operation.
*E*	Exponential calculation time.

**Table 3 entropy-25-01334-t003:** Cryptography operations’ time in milliseconds.

Cryptography Operation	$T_{{\mathrm{SM}}_{E}}$	$T_{A_{E}}$	$T_{H_{P}}$	$T_{M}$	$T_{P}$	$T_{E}$
**Execution time (ms) **	1.7090	0.0075	4.406	0.042	5.071	8.31

**Table 4 entropy-25-01334-t004:** Efficiency analysis of ring signature algorithms.

Algorithm	Signature Generation	Signature Verification
[27]	$6nT_{P}+4nT_{E}$	$nT_{E}+2nT_{P}$
[28]	$\left(2n-1\right)T_{M}+4nT_{E}$	$2nT_{E}+2TP$
[25]	$\left(4n-1\right)T_{M}+\left(4n+6\right)T_{E}$	$nT_{E}+2T_{P}$
Ours	$\left(4n-2\right)T_{SM_{E}}+\left(2n-1\right)T_{H_{P}}+T_{M}$	$4nT_{SM_{E}}$

## Data Availability

Not applicable.

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
