# Peer review of "Secure Ring Signature Scheme for Privacy-Preserving Blockchain"

_entropy, 2023, doi:10.3390/e25091334_

Round 1

Reviewer 1 Report

Identity linkability is an interesting problem in transparent blockchains. This paper proposed a secure ring signature scheme for mitigating this problem by combining distributed key generation (DKG) and elliptic curve cryptography (ECC). Theoretical analysis shows correctness, unforgeability, unconditional anonymity, and complexity. However, some issues should be considered.

1. The contributions are limited. It seems that the proposed scheme is generic to most ECC-based applications instead of blockchains. Also, this paper did not consider the specific requirements of blockchains.

2. The novelty is not enough. The proposed ring signature scheme is only a direct combination of existing DKG and ECC schemes. There is no technical challenge in designing and deploying the proposed scheme.

3. The introduction should contain the main contributions and organization.

4. This paper is short of the introduction to the system model. A system model figure may be helpful for readers to understand.

5. In the detailed construction, the proposed scheme requires a centralized KGC, which is conflicted with the assumption in most DKG-based blockchains.

6. There are many typos in the paper, such as “cryptography, etc., it has the” in the abstract and “efficiency of signature” in the conclusion. Please proofread the paper.

Some Grammar mistakes should be resolved. 

Author Response

Dear Editor and Reviewers,

We would like to sincerely thank you for handling the review process of our paper and recognizing our research work. We are also indebted to you and the reviewers for the valuable comments that helped us greatly improve the quality of the manuscript.

We have addressed all the reviewers’ comments and highlighted the revised parts in red in the revised manuscript. And we have our point-by-point responses in the following. In this response letter, all the reviewers’ comments are typeset in italic font and our responses are written in plain font. Rephrased sentences are typeset in red. Page numbers refer to the single column version of the revised manuscript.

Yours Sincerely,

Lin Wang

Comments by Reviewer 1

  1. The contributions are limited. It seems that the proposed scheme is generic to most ECC-based applications instead of blockchains. Also, this paper did not consider the specific requirements of blockchains.

Response: Thank you for your careful comments. Most of the existing ring signature schemes suitable for blockchain are based on bilinear pairing construction, and do not consider the risk of key exposure faced by a central key generation center. Therefore, we propose a new signature scheme. We re-describe the combination process of signature and blockchain in detail in Section 4 of the paper and give the corresponding system model diagram.

  1. The novelty is not enough. The proposed ring signature scheme is only a direct combination of existing DKG and ECC schemes. There is no technical challenge in designing and deploying the proposed scheme.

Response: Thank you for your careful comments. The proposed scheme in this paper is a redesign of ring signature, which is different from the technical principle of the traditional ring signature scheme, combining DKG, ECC and blockchain, the new scheme has higher security and better efficiency compared with other related schemes. In this revision, we have added a system model and a detailed description of the deployment aspects of the blockchain network operation.

  1. The introduction should contain the main contributions and organization.

Response: Thank you for your careful comments. We have re-revised the introduction section as requested.

  1. This paper is short of the introduction to the system model. A system model figure may be helpful for readers to understand.

Response: Thank you for your careful comments. We have added the system model and response description in Section IV.

  1. In the detailed construction, the proposed scheme requires a centralized KGC, which is conflicted with the assumption in most DKG-based blockchains.

Response: Thank you for your careful comments. The KGC proposed in this paper generates Qi based on the user's identity information to verify and hide the user's identity, and does not directly generate the user key, and then the user generates the corresponding key pair for subsequent signatures based on Qi and the system master secret key by their own calculations, and the system's master secret key is generated by the group of users in a distributed manner, so that the KGC proposed in this paper is not in conflict with the blockchain assumptions of the DKG. Since there is an understanding ambiguity in the KGC description scheme, this paper changes KGC to KDC and explains the explanation.

  1. There are many typos in the paper, such as “cryptography, etc., it has the” in the abstract and “efficiency of signature” in the conclusion. Please proofread the paper.

Response: Thank you for your careful comments. We have carefully checked the typos and finished correcting them.

Reviewer 2 Report

This article focuses on an interesting and topical subject. Moreover, the description of the proposed solution, although short, is clear. However, a number of points need to be considered before a publication of this work can be envisaged:

1) "which overcomes the issues of poor performance, unreliable data storage and high dependence on the centralized system": is this really one of the advantages of blockchain? Existing Blockchain solutions don't seem to offer much performance gain over conventional systems, do they?

2) The organization of the introduction and the article in general is quite unusual. For example, the introduction is purely descriptive, and as it stands does not include either the positioning of this article (limits of the existing) or its contributions. This should be reworked to make the article clearer and more relevant. The Related Works section does not have to be part of the initial positioning of the article.

3) In this same Related Works section, the number of articles described is very large, but the actual description of the article and presentation of their differences/benefits remains very limited. Perhaps the use of a table could be relevant to help the reader identify the contribution/limitations of the existing and the positioning of the proposed solution? What's more, the use of distributed key generation seems to be an idea that has already been exploited for master key generation, doesn't it? What are the real benefits of the proposed solution?

4) Sections 5 and 6 remain extremely limited. How can we assess the relevance of the solution in the absence of implementation and comparison with other existing solutions? I find it difficult even to consider section 6 as it stands as an evaluation section.

5) From the point of view of form, a convention should be adopted for the use of acronyms (capital letter or not at the time of definition). In addition, I think that the explanation of certain acronyms is lacking: VANET, P2P, etc.

Some sentences could be reworked, but the average level is more than acceptable.

Author Response

Dear Editor and Reviewers,

We would like to sincerely thank you for handling the review process of our paper and recognizing our research work. We are also indebted to you and the reviewers for the valuable comments that helped us greatly improve the quality of the manuscript.

We have addressed all the reviewers’ comments and highlighted the revised parts in red in the revised manuscript. And we have our point-by-point responses in the following. In this response letter, all the reviewers’ comments are typeset in italic font and our responses are written in plain font. Rephrased sentences are typeset in red. Page numbers refer to the single column version of the revised manuscript.

Yours Sincerely,

Lin Wang

Comments by Reviewer 2

  1. "which overcomes the issues of poor performance, unreliable data storage and high dependence on the centralized system": is this really one of the advantages of blockchain? Existing Blockchain solutions don't seem to offer much performance gain over conventional systems, do they?

 Response: Thank you for your careful comments. Performance is not a blockchain advantage, the expression of this article is wrong and has been deleted. The traditional centralized data system has problems such as data silos and information are easy to be tampered with, blockchain as a distributed storage system, the uploaded information needs to be verified by all nodes, and the information records that have been uploaded are open and transparent, so that no one or organization can tamper with them, and the security of the information is guaranteed. So, for the shortcomings of traditional data storage solutions, blockchain system can really improve these problems. However, due to the characteristics of blockchain, it suffers from poor scalability, low efficiency, and privacy leakage. The proposed scheme in this paper focuses on how to protect user identity privacy in blockchain system.

  1. The organization of the introduction and the article in general is quite unusual. For example, the introduction is purely descriptive, and as it stands does not include either the positioning of this article (limits of the existing) or its contributions. This should be reworked to make the article clearer and more relevant. The Related Works section does not have to be part of the initial positioning of the article.

Response: Thank you for your careful comments. We have reshuffled the logic of the introduction, adding article contributions and structure as required.

  1. In this same Related Works section, the number of articles described is very large, but the actual description of the article and presentation of their differences/benefits remains very limited. Perhaps the use of a table could be relevant to help the reader identify the contribution/limitations of the existing and the positioning of the proposed solution? What's more, the use of distributed key generation seems to be an idea that has already been exploited for master key generation, doesn't it? What are the real benefits of the proposed solution?

Response: Thank you for your careful comments. The articles in the "Related Works" section have been streamlined as required, and the different signature schemes mentioned are summarized and compared in a table.

  1. Sections 5 and 6 remain extremely limited. How can we assess the relevance of the solution in the absence of implementation and comparison with other existing solutions? I find it difficult even to consider section 6 as it stands as an evaluation section.

Response: Thank you for your careful comments. We have revised the article as requested. Signature schemes have been reevaluated according to the experimental environment, while multiple signature schemes have been added for efficiency comparison.

  1. From the point of view of form, a convention should be adopted for the use of acronyms (capital letter or not at the time of definition). In addition, I think that the explanation of certain acronyms is lacking: VANET, P2P, etc.

Response: Thank you for your careful comments. For ease of understanding, we have explained the abbreviations in the article.

Reviewer 3 Report

1. The introduction needs to be rewritten, it consists of repeating the same idea many times. Also, in the introduction, it needs to be explained what will be done in the work, why this research is needed.

2. Related works consist of fairly old papers, mostly 2013–2015; please explain the state of the art nowadays, and the section with explanation what will be done in the work is not suitable for "related works"

3. Section 4—main section, please make it more readable and understandable; now it is quite difficult to follow.

4. Please check the text for typos and mistakes. For example, line 360

5. The conclusion needs to be improved, now it looks very bad.

Author Response

Dear Editor and Reviewers,

We would like to sincerely thank you for handling the review process of our paper and recognizing our research work. We are also indebted to you and the reviewers for the valuable comments that helped us greatly improve the quality of the manuscript.

We have addressed all the reviewers’ comments and highlighted the revised parts in red in the revised manuscript. And we have our point-by-point responses in the following. In this response letter, all the reviewers’ comments are typeset in italic font and our responses are written in plain font. Rephrased sentences are typeset in red. Page numbers refer to the single column version of the revised manuscript.

Yours Sincerely,

Lin Wang

Comments by Reviewer 3

  1. The introduction needs to be rewritten, it consists of repeating the same idea many times. Also, in the introduction, it needs to be explained what will be done in the work, why this research is needed.

Response: Thank you for your careful comments. We have rewritten the introduction as requested, reorganized the logic, simplified the paragraphs, and increased the contribution and overall structure of the article.

  1. Related works consist of fairly old papers, mostly 2013–2015; please explain the state of the art nowadays, and the section with explanation what will be done in the work is not suitable for "related works"

 Response: Thank you for your careful comments. We have removed most of the old papers from 2013 to 2015, added signature schemes from recent years, and summarized and compared the mentioned signature schemes.

  1. Section 4—main section, please make it more readable and understandable; now it is quite difficult to follow.

Response: Thank you for your careful comments. We have added the system model and response description in Section IV.

  1. Please check the text for typos and mistakes. For example, line 360

Response: Thank you for your careful comments. We have carefully checked the typos and finished correcting them.

  1. The conclusion needs to be improved, now it looks very bad.

Response: Thank you for your careful comments. We have revised the conclusion part of this manuscript. The characteristics of our scheme have been re-stated, and reflections have been made on the next steps in the research.

Round 2

Reviewer 1 Report

My questions were revised. The manuscript can be accepted in this version. 

Reviewer 2 Report

The authors have taken all the comments that were made into account, and I'd like to thank them for that. The paper is much more pleasant to read as it stands, and more complete.

The quality of the experiments could still be improved, but it may seem sufficient as it stands.

It might be a good idea, perhaps, to rewrite the first paragraph of the introduction, breaking it down into a larger set of sentences.

Reviewer 3 Report

Papper can be accepted for publication.